# Genetic Algorithm Design of MOF-based Gas Sensor Arrays for CO_2_-in-Air Sensing

**DOI:** 10.3390/s20030924

**Published:** 2020-02-10

**Authors:** Brian A. Day, Christopher E. Wilmer

**Affiliations:** 1Department of Chemical and Petroleum Engineering, University of Pittsburgh, 3700 O’Hara St, Pittsburgh, PA 15261, USA; brd84@pitt.edu; 2Department of Electrical and Computer Engineering, University of Pittsburgh, 3700 O’Hara St, Pittsburgh, PA 15261, USA

**Keywords:** metal-organic framework, computational design, carbon dioxide, gas sensing

## Abstract

Gas sensor arrays, also known as electronic noses, leverage a diverse set of materials to identify the components of complex gas mixtures. Metal-organic frameworks (MOFs) have emerged as promising materials for electronic noses due to their high-surface areas and chemical as well as structural tunability. Using our recently reported genetic algorithm design approach, we examined a set of 50 MOFs and searched through over 1.125 × 10^15^ unique array combinations to identify optimal arrays for the detection of CO_2_ in air. We found that despite individual MOFs having lower selectivity for O_2_ or N_2_ relative to CO_2_, intelligently selecting the right combinations of MOFs enables accurate prediction of the concentrations of all components in the mixture (i.e., CO_2_, O_2_, N_2_). We also analyzed the physical properties of the elements in the arrays to develop an intuition for improving array design. Notably, we found that an array whose MOFs have diversity in their volumetric surface areas has improved sensing. Consistent with this observation, we found that the best arrays consistently had greater structural diversity (e.g., pore sizes, void fractions, and surface areas) than the worst arrays.

## 1. Introduction

There has been continued interest in addressing the challenge of recreating the gas-sensing capabilities of a biological nose via a gas-sensing array, known as an electronic nose [1,2,3]. Unlike other gas-sensing technologies, such as gas chromatography (GC) or mass spectroscopy (MS), which can be slow, expensive, and are largely not portable, an electronic nose can be small, cheap, and provide real-time sensing [1,4]. However, designing an electronic nose that can rival a biological nose is enormously difficult, in part because gas mixtures in ambient environments are highly complex, often including hundreds of species that competitively bind to the same receptors [3]. Nevertheless, there are numerous applications where an electronic nose would not need to compete with a biological nose, but rather perform in ways where current gas-sensing technologies are insufficient, whether due to a long response time or lack of sensitivity, such as in disease detection [4,5,6,7], or due to a lack of portability, such as for emergency response [8], agriculture [6,9,10], and mobile robots [10,11,12].

One application that is seeing renewed interest is in the detection of carbon dioxide, primarily because of its role in climate change [13,14,15]. There is, consequently, an obvious motivation to be able to reliably detect and monitor levels of CO_2_ in the atmosphere. Additionally, with geologic carbon sequestration gaining traction as a viable carbon capture and storage (CCS) technology to mitigate the greenhouse gas effect of CO_2_, the need to detect leaks from storage sites grows with it [16,17,18]. Other applications include the detection of CO_2_ in confined spaces where it can rapidly accumulate, especially when tanks of CO_2_ or dry ice are being used [19]. And even in areas where CO_2_ is less likely to accumulate rapidly, there are concerns over its role as an indoor pollutant [8,20,21].

Exposure to elevated levels of CO_2_ in air poses a two-fold threat, acting as both an asphyxiant by displacing oxygen, and as a toxicant, both with potentially deadly consequences. At levels greater the 5%, it can result in the development of hypercapnia, a build-up of CO_2_ in the bloodstream, and respiratory acidosis, an inability to clear excess CO_2_ from the lungs [8,20,22]. At concentrations of 10% and greater, exposure can result in convulsions, coma, and even death. And at levels of 30% and greater, exposure can lead to a loss of consciousness in only a matter of seconds [8,20]. Clearly, rapid, sensitive, and portable (even wearable) CO_2_ sensors would benefit many people.

Recently, a number of studies have investigated the use of metal-organic frameworks (MOFs) for improving electronic noses [23,24,25,26,27]. MOFs are promising materials for gas adsorption applications due to their nanoporous nature, high internal surface areas, and ease of tunability [28,29,30,31,32,33,34]. Moreover, as crystalline materials, they lend themselves conveniently to accurate computational modeling not possible with their porous amorphous counterparts [35,36,37].

Practically, however, there are still many significant obstacles to overcome in developing MOF-based electronic noses. While tens of thousands of MOFs have been reported in the literature, only a small fraction have simultaneously been amenable to thin-film deposition and also possess the stability required for a practical device. However, even if those material-property obstacles were overcome, an import design challenge has remained largely overlooked: what combination of MOFs optimizes performance? Although it is intuitive that adding more sensing elements to an array should improve performance, to what degree the performance can be improved has not been widely explored [38]. Similarly, there has been little work on systematically finding the best combinations of MOFs for an array, and with thousands of MOFs to choose from, and thus well over 10^50^ possible arrays, determining the top performing arrays is highly non-trivial. A recent paper by Sturluson et al. made significant progress in this area by computationally designing MOF-based gas sensing arrays for the detection of CO_2_ and SO_2_ in dilute conditions [27]. In their paper, they calculated the Henry’s coefficients, which are used to predict adsorption of gases in dilute concentrations, for each gas/MOF to design arrays with maximal complementarity, and consequently the best signal.

In this paper, we use a computational methodology to find optimal MOF-based gas-sensing arrays for CO_2_ detection in ambient air, where the MOFs are assumed to be deposited as thin films on sensors that detect changes in mass, such as surface acoustic wave (SAW) devices, as shown in Figure 1, or quartz crystal microbalances (QCMs). The number of possible arrays one could construct grows as *N*! where *N* is the number of MOFs to choose from in a given library (the size of the search space also depends on the maximum array size allowed, see the Appendix A
Section 3.2 for more details). In our case, from a library of 50 MOFs where arrays could have any size from 1 through to 50 MOFs, one could construct over 1.125 × 10^15^ unique arrays. Since a brute force search clearly could not be used for such a large space, we used a genetic algorithm strategy to quickly identify high performing arrays of different sizes, while only exploring a fraction of the space. Genetic algorithm searches were constrained to fixed time (i.e., fixed number of arrays considered) and so the computational cost was independent of the size of the search space. With the resulting data, we examined the physical properties of the MOFs in each array to find which combinations of properties are typically found in the best (or worst) arrays for CO_2_-in-air sensing.

## 2. Materials and Methods

To model CO_2_ in air, we studied a set of ternary gas mixtures containing CO_2_, O_2_, and N_2_. The compositions of CO_2_ and O_2_ ranged from 0% to 30%, and the composition of N_2_ ranged from 40% to 100%, each in increments of 1%. This resulted in 961 unique gas mixtures.

Using the molecular modeling software RASPA [39], we ran grand canonical Monte Carlo (GCMC) simulations to calculate the adsorption of each gas mixture in a set of 50 MOFs at a temperature of 298 K and a pressure of 1 bar, reflecting ambient conditions. We used the same set of 50 MOFs as was used in a prior work on methane-in-air sensing [24], which were selected from the CoRE MOF database [40] with the aim of having a diverse representation of surface areas and void fractions.

To model electrostatic interactions, which are important for accurately predicting CO_2_ and, to a lesser extent, N_2_ adsorption, we assigned partial charges to the atoms of the MOF frameworks via the EQeq method [41]. Similarly, the molecule parameters of the gases also included partial charges, and the forcefield which we used, TrAPPE, has been shown to accurately simulate these effects [42].

Once we had a complete set of data from the adsorption simulations, the next step was to design and evaluate gas sensing arrays with the elements, emulating SAW sensors that have had MOF thin films deposited on them. SAW sensors measure changes in the mass of these thin films due to gas adsorption. To recreate this type of signal, we used a subset of the simulated adsorption values as the sensor outputs here, those values coming from simulations of 5% CO_2_, 20% O_2_, and 75% N_2_. By comparing the sensor output from each array element to our library of simulated data, we can assign a probability to each of the 961 possible gas mixtures.

Note that since sensor outputs are simply a subset of the simulated data, there is always a composition whose simulated adsorption values are a perfect match. However, since there is always some error in measurements from real sensor devices, we account for this by artificially imposing measurement error on the sensor output, which ensures that no gas mixture can be predicted with 100% certainty.

An overview of the process of assigning these probabilities is as follows: one MOF at a time, we take the output signal (i.e., mass) associated with that MOF and create a truncated normal probability curve centered about the mass, with standard deviation 5% of that mass. The intention of using a truncated probability distribution rather than a true normal distribution is to account for the fact that adsorption will always result in an increase in mass. Consequently, the lower bound is set at 0, and the upper bound is set far beyond the highest simulated mass present in the dataset.

For each composition, we assign a probability based on where simulated mass sits on the truncated probability curve. Since each composition is assigned a probability independent of each other, the sum of all probabilities does not necessarily equal 1. However, since the intention of this process is to determine which simulated composition we have exposed the array to, we normalize the assigned probabilities for each MOF so that now their sum equals 1. This process is repeated for each MOF until we have one normalized probability value for each composition for each MOF.

Now that we have the probabilities for each MOF, we need to determine the probabilities for arrays. Fortunately, this process is straightforward. For each composition, we simply multiply all of the normalized probabilities for each MOF with each other, resulting in a non-normalized array probability for each composition. As before, we normalize these probabilities so that they sum to 1. With this information, we can now say which of the simulated gas mixtures best matches the sensor output of any given array. For a more complete description about this process, please refer to the Appendix A, as well as to the previous work which this is building upon [24,25,26].

In order to quantify the array performance beyond simply looking at the probability of the gas mixture we are testing for, we used a metric known as the Kullbeck-Liebler Divergence (KLD) [43], which is calculated (in units of bits) as follows:(1)$$\mathrm{KLD}=\overset{N}{\underset{i=1}{\sum }}P_{i}\cdot \log_{2}\left(P_{i}\cdot N\right)$$
where P_i_ is the probability of composition i and N is the total number of possible compositions. In essence, the KLD tells us how much better we predict a composition over random chance, with a higher KLD corresponding to better prediction.

One critical advantage of the KLD is that it is calculated without any knowledge of the experimental gas mixture; that is to say, it is not an assessment of the accuracy of the prediction, but rather an assessment of the uniqueness of the signal. Furthermore, it is a metric that is transferrable to real sensor devices, not just the computational devices we examine in this paper. As such, the KLD score could even be used as a metric for determining the minimum performance requirements for a given array. Additional details on the KLD score are given in the Appendix A.

Lastly, for designing and screening arrays, we had two approaches available to us: a brute force approach, in which we evaluate all possible arrays of a given size, and a genetic algorithm approach [24], in which we continually update a set of arrays with a mutation strategy in order to seek the best performing arrays. Even with only 50 MOFs to choose from, there are over 2.1 × 10^6^ possible 5-element arrays. For 25-element arrays, the number of possibilities grows dramatically to over 1.25 × 10^14^ arrays; hence, the need for the genetic algorithm. Although many other search algorithms could also have been used, we chose to use a genetic algorithm for its relative simplicity, easy implementation, and broad familiarity in the scientific community. Additionally, since the objective function we chose for the genetic algorithm simply seeks to maximize the KLD, it makes no implicit assumptions about which features of an array are desirable and can thus be used to develop intuition for array design, as is done in this work. A detailed description of the genetic algorithm approach can also be found in the Appendix A.

## 3. Results

### 3.1. Complementary Effect of Sensing Elements in a Single Array

As a baseline for understanding how well the MOFs we had selected perform for CO_2_ sensing, we examined the performance of all 1-element arrays, which is to say we examined how well each of the MOFs can individually predict the composition as a stand-alone sensor. Given the mixture of CO_2_, O_2_, and N_2_, an interesting result appears; all of the MOFs studied show a unique sensitivity to CO_2_ over the other two gases (see Figure 2a–c). This is likely because CO_2_ has a significant quadrupole moment and adsorbs more strongly than either O_2_ or N_2_. Figure 2a–c shows, from all 50 of the 1-element arrays, the best performing MOF (based on its KLD score) was Mg-MOF-74 [44], the 25th best was Cu_4_I_4_(DABCO)_2_ MOF [45], and the worst was the La(PODC)_1.5_(H_2_O) MOF [46].

Given that each MOF showed selective binding for CO_2_ (over either O_2_ or N_2_), one might guess that the possibility for finding complementary combinations of MOFs to detect O_2_ and N_2_ would be impossible. Fortunately, even seemingly small differences in the adsorption behavior of the individual MOFs has a way of resolving itself into improved predictions once the arrays are large enough (see Figure 2d–f). By adding a handful of (carefully chosen) elements, we are able to narrow our prediction of all components down dramatically, and adding even more elements enables us to further improve our confidence. Figure 2d–f shows the best performing 5-, 15-, and 25- element arrays that illustrate this effect. Using only (the best-found) 5-element array, we were able to accurately predict the composition with 0.64% certainty, relative to the other 960 possible compositions; by 25-elements, we were able to accurately predict the composition of the gas with 39.3% certainty (i.e., a 39.3% chance of landing on the correct composition, 5% CO_2_, 20% O_2_, and 75% N_2_, instead of any of the other 960 possible compositions).

### 3.2. Brute Force and Genetic Algorithm Results

As a test of the genetic algorithm, we calculated by brute force all 1-, 2-, 3-, 4-, and 5-element arrays, and then screened arrays of the same size with a genetic algorithm (see Figure 3). We ran the algorithm three times seeking the best arrays, and another three times seeking the worst arrays, for a total of 6 runs per array size. Each run of the genetic algorithm included 20 arrays per generation and 200 generations per run. As shown in Figure 3, the genetic algorithm is clearly successful in finding the highest performing arrays.

### 3.3. Maximum KLD vs. Number of Elements in an Array

Another key advantage of the KLD metric is that its range of values is limited only by the number of possible outcomes, so in the case of gas sensing, the number of possible compositions. As such, it can be used to compare arrays independent of size, individual sensing elements, or even the type of sensing mechanism, so long as the number of possible compositions is the same. Consequently, we can use the KLD value to show the advantage of adding more elements to a sensing array, or conversely, the diminishing returns of adding too many elements.

Figure 4 shows the best and worst KLD values of the arrays by size. Only arrays of size 1−5 were done using the brute force method; all other array sizes were screened via the genetic algorithm, hence the best, worst, and average KLD is that of all the screened arrays, not of all possible arrays.

It is clear that for smaller arrays, the overall performance is worse. The maximum achieved KLD for a 1-element sensor is only 3.46 bits, while for a two-element sensor, it is 3.97 bits, and for a three-element sensor, it is 4.35 bits. By 45 elements, the maximum-achieved KLD is 7.01 bits. It also becomes apparent from Figure 4 that the range of possible KLD values is much greater for small arrays than for large ones. Although the decrease in variability of the KLD for large arrays moderates the need for advanced screening approaches like the genetic algorithm, it underscores the advantages of employing many-element sensors over 1-element sensors.

## 4. Discussion

Identifying combinations of physical properties that lead to high performance sensor arrays could give insights on how to build sensing arrays that use MOFs outside of the 50 specifically considered in this study. Thus, here we examine the combinations of physical properties of the MOFs in the best and worst performing arrays. Intuitively, one might expect that an array made up of a diverse set of materials would perform better than an array of nearly identical materials. Conversely, it has been shown that arrays with similar materials do better at handling noise [38].

Although there are many different physical properties one could choose to examine, there are a few that are particularly likely to be important based on known trends in gas adsorption. At low pressures, adsorption behavior is typically governed by the heat of adsorption. The relative heats of adsorption between different components also provides insight into selectivity, which could be beneficial for sensing applications [35,47,48]. At moderate pressures, adsorption behavior has been shown to correlate strongly with surface area [35,47,48,49]. And finally, at high pressures, adsorption behavior is dominated by the free volume of the pores of the material, with gas molecules successfully packing all accessible pores and channels [35,47,48,49].

Hence, we examined the influence of the following properties on gas array sensing performance: volumetric surface area, void fraction, and pore diameter. In Figure 5, the 300 best and 300 worst 5-element arrays, ranked from best to worst, are plotted against the physical properties of the elements in the array. We chose to examine 5-element arrays, rather than larger arrays, since with smaller arrays it is harder to compensate for bad elements. Furthermore, with larger arrays, we observed a narrowing of the minimum and maximum KLD values. For 5-element arrays, the KLD of the 300 best-found arrays ranges from 4.94–4.58 bits, and for the 300 worst-found arrays the KLD ranges from 2.51–1.87 bits.

With volumetric surface area, the best arrays feature elements spanning a wide range of the available surface areas, corresponding to a higher standard deviation. Conversely, the worst arrays feature elements with a narrow range of surface areas, corresponding to a lower standard deviation, indicating that for this feature, a diverse set of materials is preferred.

Similarly, the best arrays feature elements with a diverse set of void fractions. Here, however, both the best and worst arrays feature similar ranges of void fractions, and consequently the standard deviations are comparable. Nevertheless, the best arrays show more diversity over that range, whereas with the worst arrays often contain sets of elements with nearly identical void fractions, once again indicating that a diverse set of materials seems to be preferred. Additionally, the void fractions of the worst arrays are shifted higher than those for the best arrays, which is likely to lead to decreased selectivity, especially with similarly sized gases.

Consistent with before, we notice that the best arrays feature a diverse distribution of pore sizes; however, this diversity occurs over a somewhat limited range [~4–30 Å], as compared to the worst arrays [~5–45 Å]. This likely indicates that, as before, a diverse set of elements is beneficial, but that there is a limit to the usefulness of excessively small (<~4 Å) or large pores (>~30 Å), as beyond those limits they either exclude all gases or are so large that the volumetric density of chemically selective adsorption sites is too low for appreciably selective binding.

In addition to examining the complementarity of the individual physical properties of the sensing materials, we also wanted to examine whether there was complementarity across different physical properties (see Figure 6). For example, it might be desirable to have an element with a high surface area, but a narrow pore diameter, so that it selectively adsorbs only smaller gases, but in large quantities, providing a uniquely good signal for detecting those components. No complementary effect between any two properties (e.g., large surface areas complementing small pore sizes) becomes readily apparent, so it might be safe to conclude that for this gas mixture at these conditions, it is enough to consider only the individual properties of the elements. Nevertheless, there is still a noticeable clustering of the blue dots (worst array elements) compared to the red dots (best array elements), which continues to suggest that diversity of properties is beneficial for high-performing arrays.

When considering the data shown in Figure 5 and Figure 6, it is important to remember that there are only 50 MOFs to choose from, and the gas mixtures feature only three gases of similar sizes (kinetic diameter of CO_2_ is 3.30 Å, of O2 is 3.46 Å, and of N_2_ is 3.64 Å) [50]. As such, the selected MOFs may not be representative of the whole material class (though they were chosen to be relatively diverse) and the gases are clearly not representative of gases species at large. Thus, the conclusions from the above figures may not be applicable for all gas-sensing applications, but rather specific to detecting CO_2_ in air. Moreover, with CO_2_ being strongly polar (relative to N_2_ and O_2_) and more likely to exhibit strong binding relative to these gases, it is plausible that other features, such as open metal sites, have a greater influence on the adsorption behavior, and thus are better predictors of and design criteria for the best/worst arrays than the features examined here [51,52,53]. Nevertheless, our study and the methodology described herein can help guide our intuition with regards to the importance of these features for gas sensing.

Given these results, we would recommend to an experimentalist designing gas-sensing arrays for CO_2_ in air to choose MOFs with a wide variety of surface areas and void fractions. Furthermore, we would recommend avoiding MOFs whose pore sizes are much larger than size of the gases of interest, in order to achieve improved selectivity.

Given these insights, experimentalists aiming to develop a MOF-based sensing array, specifically for CO_2_ sensing, should aim for a set of MOFs with diversity of features. Namely, they should choose MOFs with a wide range of surface areas. A wide range of both void fractions and pore sizes is also useful, but one should be careful to choose MOFs over a limited scope (~0.3–0.85 for void fraction and less than ~30 Å for pore size) as MOFs with too little or too much empty space will become less selective by either adsorbing no gas or all gases, respectively.

## 5. Conclusions

Herein, we have described a methodology for screening gas sensing arrays of MOFs, specifically for the detection of CO_2_ in air. Additionally, we examined the physical properties of the MOF elements in the array to improve intuition for the design of sensing arrays.

An interesting feature of the gas mixture studied was the high sensitivity of all MOFs towards CO_2_. Although this result gave the impression that reliably detecting the O_2_ and N_2_ would be difficult, we found that with even relatively small arrays, we were able to accurately resolve their composition. Furthermore, our analysis of the physical properties of the MOFs seems to confirm our intuition that diverse arrays lead to improved sensing. A noteworthy caveat to this is that there seems to be a practical range for certain features, namely pore size, as with excessively small or large pores, selectivity is diminished by the exclusion or inclusion of all the gases present in the mixture, respectively. Although this limit is likely to change depending on the gases of interest, it seems reasonable that this concept should hold for a broad range of sensing applications.

In the future, in addition to ranking arrays over a range of compositions, we are interested in expanding the design and screening approach to include thousands of MOFs, as well as studying more complex gas mixtures, including common interferents such as carbon monoxide and water. However, this will require implementing new efficiencies to the overall procedure in order to keep computational times and costs within reason.

## Figures and Tables

**Figure 1 sensors-20-00924-f001:**
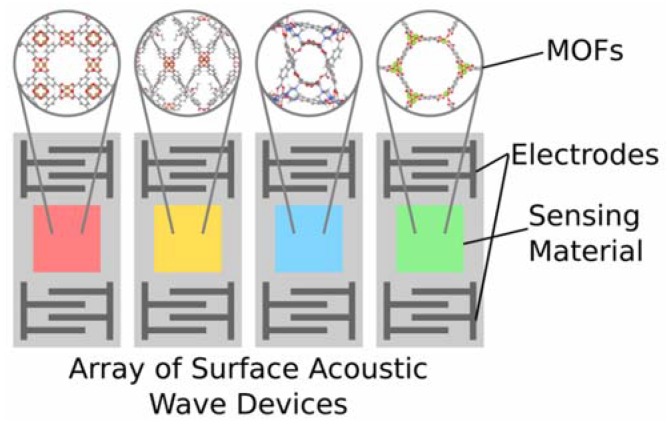
Sample four-element array of surface acoustic wave devices with MOFs as sensing materials.

**Figure 2 sensors-20-00924-f002:**
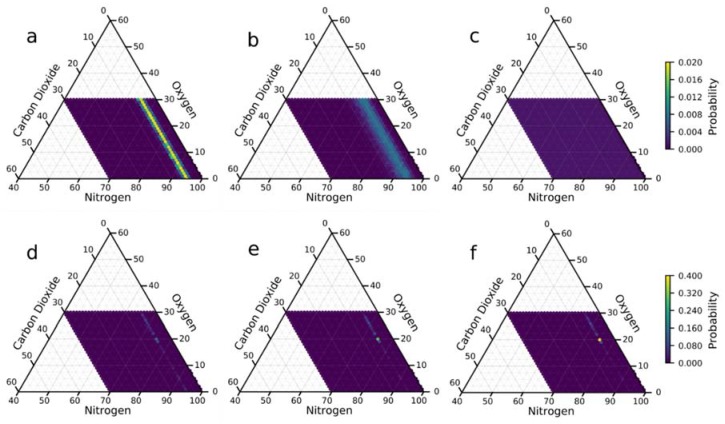
(**a**–**c**) Ternary discrete probability plots of (**a**) the best (Mg-MOF-74 [44]), (**b**) median (Cu_4_I_4_(DABCO)_2_ MOF [45]), and (**c**) worst (La(PODC)_1.5_(H_2_O) MOF [46]) 1-element arrays. (**d**–**f**) Ternary discrete probability plots of the best (**d**) 5-, (**e**) 15-, and (**f**) 25-element arrays. Please note the change in scale of the color bar for the plots in (**a**–**c**) versus the plots in (**d**–**f**).

**Figure 3 sensors-20-00924-f003:**
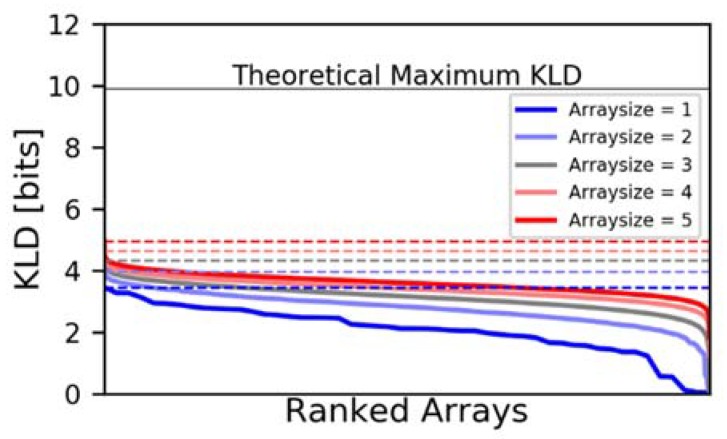
Validation of the genetic algorithm (dashed lines) against brute force screening (solid lines) for 1- to 5-element arrays. The dashed lines are the best array of each size predicted by the genetic algorithm, and the solid lines are all arrays from the brute force screening. There are more arrays as array size increases; the results are simply stretched so that the best/median/worst arrays are vertically aligned.

**Figure 4 sensors-20-00924-f004:**
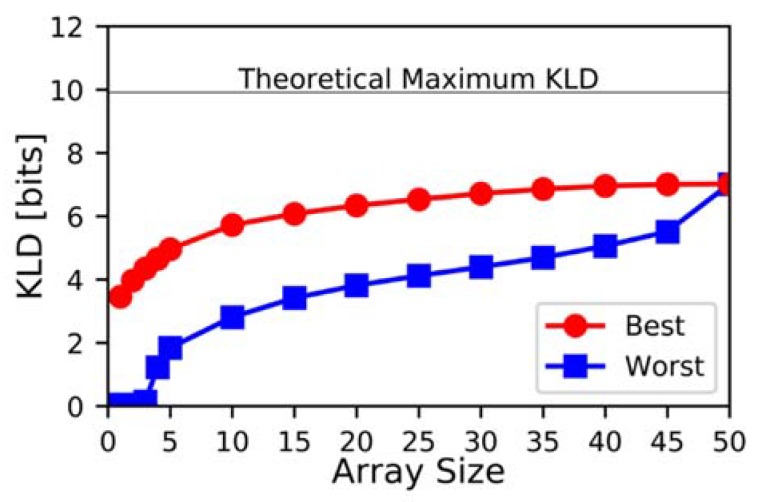
KLD in bits of best-found and worst-found as a function of the number of elements in array.

**Figure 5 sensors-20-00924-f005:**
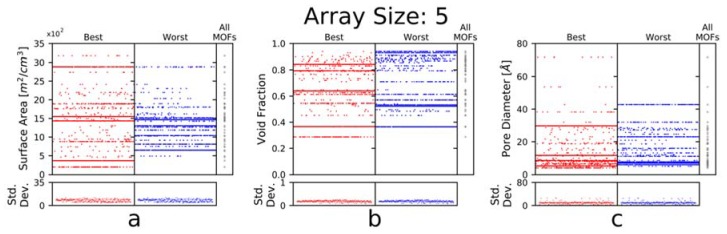
Physical property vs. rank (top/bottom 300 arrays), where (**a**) examines the surface area, (**b**) examines the void fraction, and (**c**) examines the pore diameter. The bottom portion of each plot shows the standard deviation of the corresponding physical property for each array (e.g., one array with the following five void fractions [0.79, 0.84, 0.54, 0.37, 0.64] would have a standard deviation in its void fraction of 0.17). The appearance of horizontal ‘lines’ corresponds to MOFs that are frequently featured in the best/worst arrays and gives us insight into both the (un)desirable properties as well as the (un)desirable spread of those properties.

**Figure 6 sensors-20-00924-f006:**
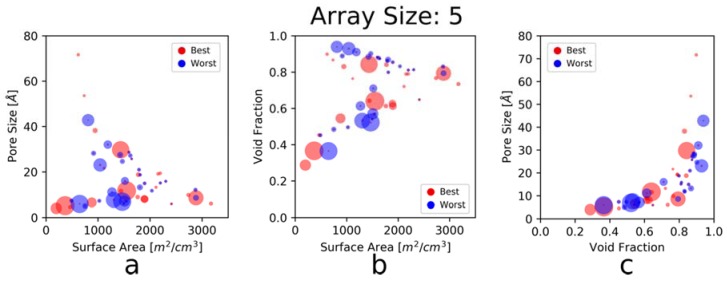
Physical property coupling (top/bottom 300 arrays), where (**a**) examines the relationship between pore size and surface area, (**b**) examines the relationship between void fraction and surface area, and (**c**) examines the relationship between pore size and void fraction. Dot color corresponds to elements of the best/worst arrays (red is for best arrays, blue for worst arrays) and dot size corresponds to the number of times the element is featured in the best/worst arrays (a larger dot corresponds to being present in more arrays). This is the same set of arrays used in Figure 5.

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
