# Peer review of "Genetic Algorithm Design of MOF-based Gas Sensor Arrays for CO2-in-Air Sensing"

_sensors, 2020, doi:10.3390/s20030924_

Round 1

Reviewer 1 Report

In the MS titled “Genetic Algorithm Design of MOF based Gas Sensor Arrays for CO2 in Air sensing” the authors have presented the use their pre-reported genetic algorithm to search for optimal set of MOFs for CO2 detection in Air. The MS can be published in Sensors after some minor revision as follows:

Since this paper is mainly about the application of genetic algorithm in finding the optimal set of MOFs, the Introduction should include literature background of other algorithms available to perform such a search and establish the advantage of genetic algorithm over other algorithms. Each term used in equation 1 should be explained in the main text. Figure 2 does not clearly establish the comparison of genetic algorithm vs brute force screening. Authors should rethink how to present the comparison data. Discussion should include a section on time complexity of the algorithm when applied to the screening

Author Response

See attached for a point by point response (all Reviewer comments included).

Reviewer 2 Report

The manuscript is well written, except for minor mistakes. The manuscript can be improved after a few clarifications on materials and methods, and by further extending the introduction and discussion. It is necessary to better explore previously reported results in the field in these sections of the text. Please, refer to the attached file for review. 

Author Response

(The authors gave the same response as above.)

Reviewer 3 Report

In this work, the authors study in simulation how well a range of CO2, O2 and N2 gas mixtures can be distinguished by using electronic noses with different combinations of metal organic frameworks (MOFs). They use a genetic algorithm to optimize the number of elements and material type in the sensor array for classifying specific mixtures against any of the other 960 possible compositions. They found that a 25-element array is able to predict the composition of the gas with 39.3% certainty, which they consider an accurate result. The paper is well structured, easy to follow and well written. My main concerns are related to the lack of important references in the Introductory section, missing important methodological details and the inclusion of some uninformative results. My comments and suggestions for improvement are outlined below:

 Intro:

  - Lines 35-37: "…perform in ways where current gas-sensing technologies are insufficient, whether due to a long response time...".  I'm not sure what the authors mean with this sentence. If current gas-sensing technologies are insufficient from a response time of view, electronic noses will be insufficient too as they are made of those gas-sensing technologies.

- Line 37: “..lack of portability, such as for emergency response.”. This is indeed an important application field of sensor arrays, especially when mounted on mobile robots or, more recently, on drones. Some suggestions for references:

* Loutfi et al. Putting olfaction into action: Using an electronic nose on a multi-sensing mobile robot. IEEE/RSJ International Conference on Intelligent Robots and Systems (IROS), 2004.

*Burgués et al. Smelling Nano Aerial Vehicle for Gas Source Localization and Mapping. Sensors. 2019.

  - The authors motivate the use of e-noses for CO2 detection, but it is well established that NDIR sensors are very convenient for this task due to its high selectivity and possibility of remote sensing. See for example:

* Pandey S, Kim KH. The relative performance of NDIR-based sensors in the near real-time analysis of CO2 in air. Sensors. 2007.

- Lines 59-60: “Although it is intuitive that adding more sensing elements to an array should improve performance, to what degree the performance can be improved has not been widely explored.”. I suggest the authors take a look at the PhD thesis of Luis Fernandez-Romero:

*Fernández Romero, L., Understanding the role of sensor diversity and redundancy to encode for chemical information in gas sensor arrays. PhD thesis. University of Barcelona. 2016

- Important references are missing regarding the use of genetic algorithms for e-nose optimization. Here are some examples:

* Gardner JW, Boilot P, Hines EL. Enhancing electronic nose performance by sensor selection using a new integer-based genetic algorithm approach. Sensors and Actuators B: Chemical. 2005.

* Kermani BG, Schiffman SS, Nagle HT. Using neural networks and genetic algorithms to enhance performance in an electronic nose. IEEE Transactions on Biomedical Engineering. 1999.

  Methods:

- The authors motivate the development of the proposed sensing system with a wide variety of applications, including measurements of atmospheric CO2 and CO2 leak detection. However, in the former case, the sensing system must be able to provide accurate and quantitative measurements of concentrations around 0.04% (CO2 concentration in outdoor air. Rather, the authors propose a classification system that can identify ternary mixtures containing CO2, O2 and N2 in different ratios. The studied range of CO2 (0-30% in steps of 1%) is so coarse that will not reveal any insight into the suitability of the proposed system for measuring atmospheric CO2 concentrations. My opinion is that the list of target applications shall be narrowed down.

- KLD metric: Why do you use this metric for evaluating the performance of a multi-class classifier instead of, for example, more standard metrics such as confusion matrix?

- Equation 1. The KLD formula shall be better explained, including the meaning of N, Pi. What is the meaning of a measurement in bits? Is it better a high KLD or a low KLD?

-The performance is calculated for only one mixture (5% CO2, 20% O2, and 75% N2) against the rest, or it is an average result of all-versus-all combinations of mixtures?

-I strongly believe that most of the information in the Suppl. Mateiral (especially Section 2) shall be included in the Methodology section of the manuscript, together with some illustrative diagram of the process described in Section 2 of the Suppl. Material.

Results:

-From Figure 3 it seems that there is not much improvement from 10 elements to 25 elements (optimum choice by the authors). I believe it is convenient to minimize the number of elements to reduce drift, power consumption, etc. How do you choose the optimum array size? This should be detailed in the Methods section.

-Figure 3 also shows that adding more elements always improves the performance, despite the authors state in line 165 that the KLD metric shows the diminishing returns of adding too many elements. If this is true and adding more elements does not suppose a big issue, then it is worthless to do any optimization. Can you clarify this apparent contradiction?

-Figures 4 and 5 do not show any clear pattern between the studied variables... Then, the inclusion of these figures and associated discussion in the paper is questionable. Why didn’t you consider the material of the sensing layer as another variable to compare against the surface area, pore size, etc.? It seems obvious that the sensing material may play a big role in the system performance.

-Do you have an estimate of the power consumption of the 25-element array?

-Do you have any plans on building the optimized array and test it against real gas mixtures?

Typos:

- Line 29: "othe1–3r".

Author Response

(The authors gave the same response as above.)

Round 2

Reviewer 3 Report

The authors have addressed most of my comments. Congratulations.